# Vagus Nerve Visualization Using Fused Images of 3D-CT Angiography and MRI as Preoperative Evaluation for Vagus Nerve Stimulation

**DOI:** 10.3390/brainsci13030396

**Published:** 2023-02-25

**Authors:** Shunsuke Nakae, Masanobu Kumon, Akio Katagata, Kazuhiro Murayama, Yuichi Hirose

**Affiliations:** 1Department of Neurosurgery, Fujita Health University, 1-98 Dengakugakubo Kutsukakecho, Toyoake 470-1192, Aichi, Japan; 2Department of Radiology, Fujita Health University Hospital, 1-98 Dengakugakubo Kutsukakecho, Toyoake 470-1192, Aichi, Japan; 3Department of Radiology, Fujita Health University, 1-98 Dengakugakubo Kutsukakecho, Toyoake 470-1192, Aichi, Japan

**Keywords:** vagus nerve visualization, VNS, MRI, 3D-CTA, fusion images

## Abstract

Vagus nerve stimulation (VNS) is an effective surgical option for intractable epilepsy. Although the surgical procedure is not so complicated, vagus nerve detection is sometimes difficult due to its anatomical variations, which may lead to surgical manipulation-associated complications. Thus, this study aimed to visualize the vagus nerve location preoperatively by fused images of three-dimensional computed tomography angiography (3D-CTA) and magnetic resonance imaging (MRI). This technique was applied to two cases. The neck 3D-CTA and MRI were performed, and the fused images were generated using the software. The vagus nerve and its anatomical relationship with the internal jugular vein (IJV) and common carotid artery were clearly visualized. The authors predicted that the vagus nerve was detected by laterally pulling the IJV according to the images. Intraoperatively, the vagus nerve was located as the authors predicted. The time of the surgery until the vagus nerve detection was <60 min in both cases. This novel radiological technique for visualizing the vagus nerve is effective to quickly detect the vagus nerve, which has anatomical variations, during the VNS.

## 1. Introduction

Vagus nerve stimulation (VNS) is a surgical option for the drug-resistant epilepsy, and its effectiveness was supported by previous studies [1,2]. Detecting the vagus nerve is occasionally difficult in some cases because of its anatomical variations although its surgical procedure is not so complicated. A previous study regarding the VNS reported the anatomical relationship between the vagus nerve and major blood vessels, such as the internal jugular vein (IVJ) and common carotid artery (CCA) [3]. Additionally, the location of the vagus nerve is classified into five patterns, and the most difficult pattern from the point of surgical manipulations is the pattern in which the vagus nerve is located dorsal to the CCA due to the deep surgical field and difficulty in vagus nerve detection. Difficulty in vagus nerve detection may lead to surgical complications, thus, understanding the anatomical vagus nerve location in advance is important to avoid complications associated with surgical manipulations.

To our best knowledge, there is an only VNS study reported on detecting vagus nerve location preoperatively using cervical ultrasonography [3]. The present study fused the images of the neck magnetic resonance imaging (MRI) and the three-dimensional computed tomography angiography (3D-CTA) for IVJ and CCA around the neck, successfully visualized the vagus nerve location preoperatively, and reports the usefulness for detecting the vagus nerve location for VNS.

## 2. Case Presentation

### 2.1. Patients

Two cases of VNS were performed using this radiological technique, including a 58-year-old female and a 53-year-old male patient as cases 1 and 2, respectively. Case 1 underwent intracerebral hematoma evacuation in the right hemisphere due to right transverse and sigmoid sinus occlusion followed by decompressive craniotomy and cranioplasty. The onset of the seizure was soon after the patient experienced the intracranial hemorrhage, thus an anti-epileptic drug (AED) was started. She experienced focal awareness seizures (FAS) a couple of times per week under three AED administrations (levetiracetam at 2000 mg, lacosamide at 400 mg, and perampanel at 4 mg) before the epilepsy surgery. Case 2 underwent craniotomy due to craniopharyngioma when he was 16 years old. The right frontal lobe was used as the surgical corridor for tumor resection at that time. Radiotherapy was performed after the tumor resection. The seizure occurred after he underwent tumor resection. Additionally, he experienced oligodendroglioma tumor resection located in the left frontal lobe when he was 44 years old. The patient experienced FAS weekly or monthly despite three AED administrations (levetiracetam at 1000 mg, lacosamide at 400 mg, and perampanel at 8 mg), and the video-electroencephalography showed that the seizures separately occurred from the right and left frontal lobes.

### 2.2. MRI Protocol

All magnetic resonance (MR) examinations were performed with a 3-T MR system (Vantage Centurian, Canon Medical Systems Corporation, Otawara, Japan) using the Atlas SPEEDER Head/Neck coil. Peripheral nerve imaging was acquired on the axial plane using three-dimensional steady-state free precession (3D-SSFP) and 3D-SSFP with time-spatial labeling inversion pulse (time-SLIP) sequences that ranged from the lower jaw to the upper clavicle. The 3D-SSFP sequence was acquired using the following parameters: TR/TE: 6.5 ms/2.5 ms; flip angle: 60 degrees; acquisition matrix: 256 × 256; reconstruction matrix: 512 × 512; acquisition resolution: 0.78 × 0.78 mm^2^; reconstruction resolution: 0.39 × 0.39 mm^2^; number of slices: 90; reduction factor: 2.5; field of view, 200 × 200 mm^2^; slice thickness: 1 mm; interslice gap: 0 mm; NEX: 2; acquisition time: 365 s. 3D-SSFP with the time-SLIP sequence was acquired using the following parameters: TR/TE: 4.4 ms/2.2 ms; flip angle: 120 degrees; acquisition matrix: 200 × 208; reconstruction matrix: 400 × 416; acquisition resolution: 1.0 × 1.0 mm^2^; reconstruction resolution: 0.5 × 0.5 mm^2^; the number of slices: 80; reduction factor: 2; black blood inversion time: 750 ms; field of view: 200 × 208 mm^2^; slice thickness: 1.5 mm; interslice gap: 0 mm; NEX: 1; acquisition time: 231 s.

### 2.3. 3D-CTA Protocol

All 3D-CTA examinations were performed with a 160-detector-row computed tomography (CT) scanner (Aquilion Precision, Canon Medical Systems Corporation, Ōtawara, Tochigi, Japan) using single-phase scanning with helical imaging. First, we performed a volumetric CT scan of the ranges, as well as MRI protocol, using the following parameters: 160 × 0.25 mm collimation; tube voltage: 120 kV; CT auto-exposure control, standard deviation: 13.0; focal spot size of the X-ray tube, the minimum (0.4 × 0.5 mm); gantry rotation time: 0.75 s/rotation; matrix: 512 × 512; field of view: 200 mm; slice thickness: 0.25 mm; helical pitch: HP91; acquisition time: 8.74 s. We used a dual head power injector (Dual Shot GX; Nemoto Kyorindo, Tokyo, Japan) in all patients to administer a bolus of iodinated contrast material (iopamidol of 250 mg/kg body weight; Iopamiron 370, Bayer Healthcare) through a cubital vein for 18 s, followed by a saline solution (30 mL) at the same rate. All CTA images were reconstructed with deep learning reconstruction (Advanced intelligent Clear-IQ Engine: AiCE, Canon Medical Systems Corporation). CTA data was acquired in the non-contrast, arterial, and venous phases using the bolus tracking technique, with a trigger at 150 HU. All MRI and CT images were transferred to commercially available software (Zaiostation 2, Ziosoft, Tokyo, Japan), and fused images were generated with semi-automatic registration by a single radiographer (A.K). In detail, using the CT images as a reference, the MRI images were manually aligned to the CT images in three dimensions. In this process, rigid registration is performed three-dimensionally using each cross-sectional (axial/coronal/sagittal) image with reference to bones and blood vessels.

### 2.4. Surgical Procedures for VNS

NIM TriVantage EMG tube (Medtronic Xomed. Inc., Jacksonville, FL, USA) was used for intubation to confirm the vagus nerve. The patient was placed in the supine position. Cervical ultrasonography was performed after general anesthesia was induced. The head was fixed in median and vertex-down positions so that neck muscles, such as a sternocleidomastoid muscle, were not loosened. The neck skin on the left side was linearly incised with 30–35 mm long at the cricoid cartilage level not to involve the cardiac branches of the vagus nerve [3,4]. The platysma muscle was transversely cut after the skin incision. The sternocleidomastoid muscle was laterally pulled, the omohyoid and sternohyoid muscles were medially pulled, and the carotid triangle was consequently exposed. Then, the CCA and IJV were identified. The vagus nerve was detected and was stimulated and confirmed using a NIM^®^ monitor after laterally pulling the IJV. Electrodes were placed around the vagus nerve after enough vagus nerve exposure, and the generator (LivaNova USA, Inc.) was placed under the skin near the left axilla. The lead electrode and the generator were connected, and the device system was checked whether or not it worked correctly.

### 2.5. Results

The vagus nerve was successfully visualized in fusion images in both cases using this method (Figure 1 and Figure 2). Unfortunately, the preoperative fused images could not discriminate the cardiac branch of the vagus nerve from the main trunk of the vagus nerve. The IJV was bifurcated at the cricoid cartilage level according to the images in case 1, and the transverse skin incision was set one finger width below the cricoid cartilage. We preoperatively predicted the vagus nerve detection by pulling the IJV laterally in both cases 1 and 2 based on the preoperative fusion images. Cervical ultrasonography was also performed just after general anesthesia was induced in both cases, and these images were revealed in Figure 3. Accordingly, the location of vagus nerve was identified in case 1, but was unclear in case 2.

In both cases, the vagus nerve was identified after pulling the IJV (Figure 4). The vagus nerve was confirmed by direct stimulation using the NIM TriVantage EMG. The time from the beginning of the surgery until vagus nerve detection was 47 min and 54 min in cases 1 and 2, respectively. In both cases, the surgery was completed without any complications.

## 3. Discussion

This study successfully visualized the vagus nerve location using the fusion images of 3D-CTA and MRI as preoperative VNS evaluation. To our best knowledge, this is the first report discussing preoperative vagus nerve detection using 3D-CTA and MRI. The vagus nerve was intraoperatively identified as we predicted in both cases using the novel techniques for two cases.

Some studies reported anatomical vagus nerve variations [5,6,7]. The ultrasonographical study demonstrated that vagus nerve variation is seen in 5.5% of 326 cases, and the vagus nerve runs dorsal to the CCA after it changes the course in 1.2% of cases [5]. Regarding the VNS, a previous study reported that the anatomical relationships between the vagus nerve and the blood vessels, including the IJV and the CCA, are classified into five patterns [3]. Accordingly, a type in which the vagus nerve is located dorsal to the CCA was seen in 4.1% of 73 cases. The authors described that pulling the CCA will be required in this type. Vagus nerve detection that is dorsally located to the CCA will be difficult without preoperative information about the vagus nerve location. Moreover, unlike the ultrasonographical study, fused images clearly visualized the thyroid cartilage and cricoid cartilage. Therefore, this technique provides us the precise information about the anatomical vagus nerve, IJV, and CCA location at the skin incision level. Apart from anatomical variations of the vagus nerve, a previous study reported the case, in which the VNS electrodes were placed on the hypertrophic vagus nerve in a neurofibromatosis type 1 (NF1) patient [8]. Therefore, the vagus nerve visualization prior to VNS might be strongly recommended in cases of NF1 patients.

Complications concerning the VNS were reported in many previous studies, including bradycardia, peritracheal hematoma, voice cord palsy, vessel injury, pain or discomfort, Horner syndrome, infection, and hardware-related complications [4,9,10,11,12]. Some of these complications are considered to be associated with surgical manipulations. The previous study indicated that vagus nerve injury is the most frequent cranial nerve injury during the carotid endarterectomy apart from VNS [13]. Therefore, understanding the vagus nerve location preoperatively and detecting it quickly is important to reduce the risk of intraoperative complications, and this novel technique can contribute to quick vagus nerve detection during surgery.

This study has some limitations. First, the motion artifact especially associated with the deglutition can be a problem during the MR examination. Shortening the imaging time by restricting the ranges of imaging may be important to avoid this problem. Second, this technique for visualizing the vagus nerve is successful in only two cases. We need to analyze more cases to show the effectiveness of this technique to detect rare anatomical vagus nerve variations. Moreover, 3D-CTA requires radiation exposure, which can be the biggest disadvantage of this technique. Because previous studies indicated that the vagus nerve location was detectable by ultrasonography [3,5,6,7], it may be considered to be enough to visualize its location. However, clinical examination of cervical ultrasonography by radiographers is usually not intended to detect the vagus nerve location or clarify the anatomical relationships around the nerve. Although we have performed cervical ultrasonography in an operating room after general anesthesia, the vagus nerve location is sometimes unclear like the image in case 2 (Figure 3). In addition, CT provides the information on the anatomical relationship between the vagus nerve and the cricoid cartilage, which is the landmark as the level of skin incision in VNS. We believe that it is also important to understand the vagus nerve location at the level of skin incision in advance. Unfortunately, we could not visualize the cardiac branches of the vagus nerve by the fusion images in the present study. Visualization of the cardiac branches should be continuously challenged as our future study because bradycardia is one of the major complications in VNS. Therefore, the present study may also serve as a base for visualizing the cardiac branches of the vagus nerve in the future.

## 4. Conclusions

The fused images of 3D-CTA and MRI preoperatively provide us the precise information about the anatomical relationships between the vagus nerve and the major blood vessels, including IJV and CCA. This novel radiological technique contributes to quick vagus nerve detection during the VNS.

## Figures and Tables

**Figure 1 brainsci-13-00396-f001:**
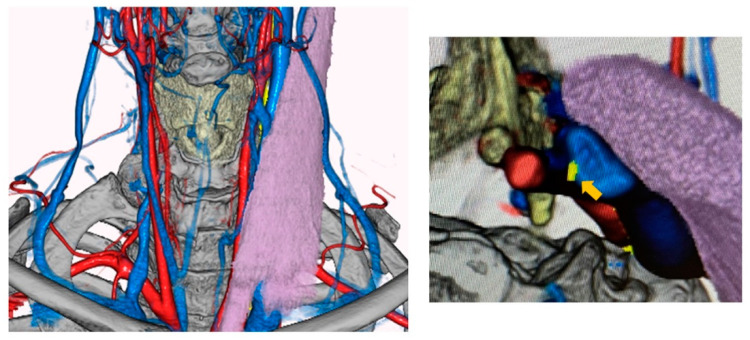
The visualized vagus nerve and its anatomical relationship with CCA and IJV in case 1. The vagus nerve was colored yellow in this figure (arrow). The sternocleidomastoid muscle, CCA, and IJV were colored purple, red, and blue, respectively.

**Figure 2 brainsci-13-00396-f002:**
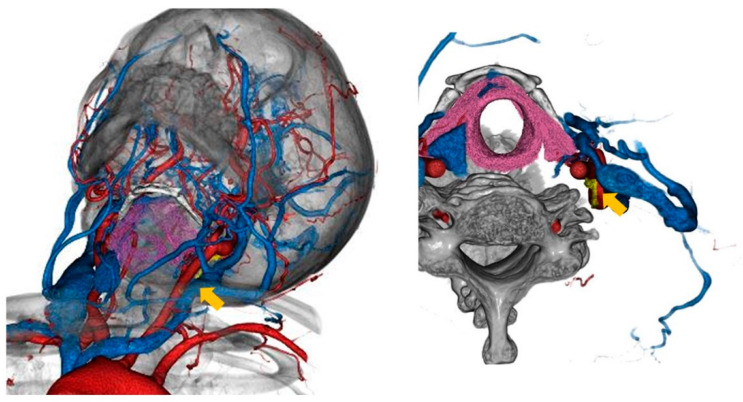
The visualized vagus nerve and its anatomical relationship with CCA and IJV in case 2. The vagus nerve was colored yellow (arrow).

**Figure 3 brainsci-13-00396-f003:**
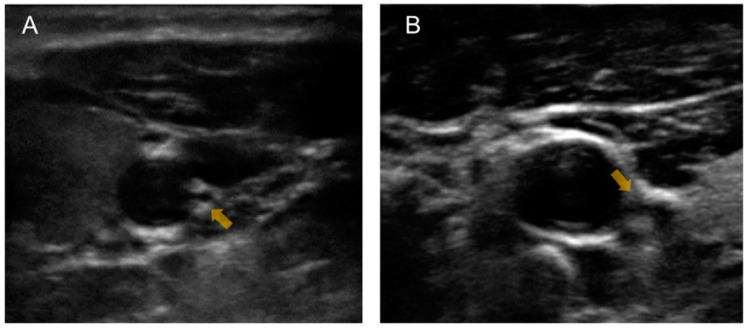
The visualized vagus nerve and its anatomical relationship with CCA and IJV by ultrasonography in case 1 (**A**) and case 2 (**B**). Although the vagus nerve was detectable in case 1, its location was unclear in case 2 (arrow).

**Figure 4 brainsci-13-00396-f004:**
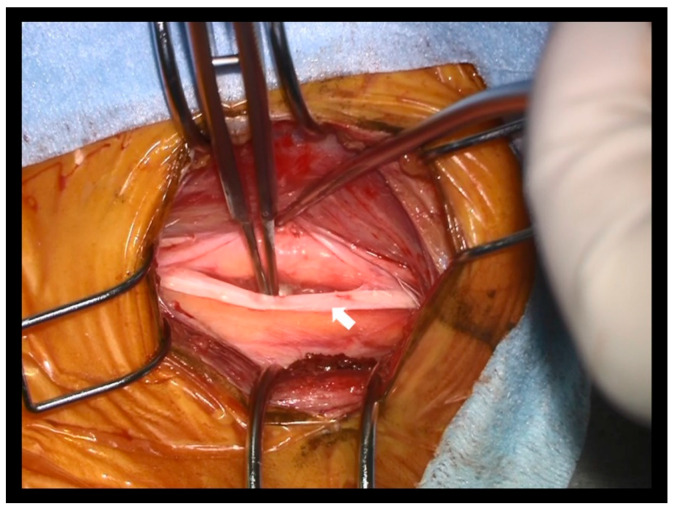
The intraoperative picture in case 1. The vagus nerve was detected by laterally pulling the IJV as the authors preoperatively predicted (arrow). The white arrow indicates the vagus nerve.

## Data Availability

Not applicable to this study.

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
