# Peer review of "Vagus Nerve Visualization Using Fused Images of 3D-CT Angiography and MRI as Preoperative Evaluation for Vagus Nerve Stimulation"

_brainsci, 2023, doi:10.3390/brainsci13030396_

Round 1

Reviewer 1 Report

This is a short report of 2 patients with epilepsy who had an implantation of vagus nerve stimulator.  The authors coregistered 3D-CT angiography and MRI images to detect the location of vagus nerve, common carotid artery and internal jugular vein in these patients. This technique was useful for avoiding the complications in the VNS implantation.

This is an interesting paper showing the usefulness of the image procesing technique for planning a surgical intervention. My only comment is that the authors should describe how to process the images in slightly more detail. The authors only say 'All MRI and CT images were transferred to commercially available software (Zaiostation 2, Ziosoft, Tokyo, Japan), and fused images were generated with semi-automatic registration by a single radiographer (A.K).', which makes the technique totally a black box. Otherwise this is a nice paper with clear purposes, results, and future possibilities.

Author Response

Thank you very much for you comments.

We added the explanation how to generate the fused images.

Reviewer 2 Report

Dear Authors,

the manuscript consists of a good description of a new imaging technique applied to the study of the vagus nerve. Unfortunately, the limit of the paper is represented by the fact that there are only two cases described.

It is necessary to better clarify the statement (lines 38 and 39) "there is an only study reported on detecting vagus nerve location preoperatively using cervical ultrasonography". Both because it leads to underestimating the substantial literature on the sonographic study of vagus nerve (see: Abdelnaby et al. Sonographic reference values of vagus nerve: A systematic review and meta-analysis. J Clin Neurophysiol. 2022 Jan 1; 39(1): 59-71. doi: 10.1097/WNP.0000000000000856). Both because there are other contributions that also deal with the specific field of vagus nerve stimulation, such as: Inamura et al. Topographical features of the vagal nerve at the cervical level in an aging population evaluated by ultrasound. Interdisciplinary Neurosurgery 2017 (September), 9: 64-67 (https://doi.org/10.1016/j.inat.2017.03.006).

A few further comments also seem appropriate: 1) on the opportunity of a preliminary anatomical study of the vagus nerve. See, for example: Hamdi et al. VNS implantation in a NF1 patient: massive nerve hypertrophy discovered intra-operatively preventing successful electrode placement. Case reports. Acta Neurochirurgica (2020) 162:2509–2512 (https://doi.org/10.1007/s00701-020-04535-y/); 2) and therefore on the fact that problems can derive not only from anatomical variants, but also from unknown pathologies of the nerve; 3) on the advantages and disadvantages of the exposed technique in comparison with ultrasonographical studies.

The manuscript must be read again to avoid printing errors (example: "allow" in the captions of the 3 figures).

Best regards

Author Response

Thank you very much for revising our manuscript.

About the statement (lines 38 and 39) we mean that there is an only study regaridng VNS to detect its location prior to the surgery. But my sentence was confusing. It was corrected accordingly.

We appreciate to tell us a case report of hypertrophic nerve. Pathologically rare case is also important. We added the sentences in Discussion.

We discussed on the advantages and disadvanged of this novel method in fourth paragraph in Discussion. But, I added explanation as Reviewer 2 pointed.

Thank you very much for telling the printing errors.  They were corrected. 

Reviewer 3 Report

The authors presented an innovative technique to identify the vagus nerve in  this technical note involving 2 patients. In this study the authors highlighted the advantages of this technique in their operative procedures.

To demonstrate the advantages of any procedure, the involvement of only 2 patients is relatively limited. Nonetheless, as a feasibility study this number is acceptable. Technical note may represent such platform.

I would like to suggest the following to improve the understanding of this technique:

1. An experienced head and neck surgeon could easily identify the vagus nerve without having to rely on such technique. Please highlight the possible clinical situations where this is indicated as absolute and relative indications.

2. The standard use of ultrasound is understandably due to its low cost and to avoid unnecessary exposure to radiation. Please discuss the advantages and disadvantages of both techniques.

Author Response

Thank you very much for  your comment.

  1. We think that radiological evaluation of the vagus nerve location may be requred due to its anatomical variations. In cases of NF1 patients, vagus nerve is sometimes hypertrophic. In such cases, preoperative radiological evaluation may be absolute indications. I added the explanaiton in Disucussion.
  2.     We discussed on advantages and disadvantages of this technique in the fourth paragraph in Discussion. But I added the explanation.

We believe that the constructive comments from the reviewers have improved the quality of our manuscript. However, we would be happy to revise the manuscript again, if necessary. Thank you very much for your time and efforts in reviewing our manuscript.